

# Allergic rhinitis, rather than asthma, might be associated with dental caries, periodontitis, and other oral diseases in adults

Sai-Wai Ho[1,2], Ko-Huang Lue[3] and Min-Sho Ku[2,3]

[1] Department of Emergency Medicine, Chung Shan Medical University Hospital, Taichung, Taiwan
[2] School of Medicine, Chung Shan Medical University, Taichung, Taiwan
[3] Division of Allergy, Asthma and Rheumatology, Department of Pediatrics, Chung Shan Medical University Hospital, Taichung, Taiwan

Corresponding author
Min-Sho Ku, a129184@yahoo.com.tw

## ABSTRACT

**Background.** The association between asthma (AS), allergic rhinitis (AR) and oral diseases remains inconclusive in adults. AS and AR often coexist. However, studies that investigate AS, AR together and their association with oral diseases are scarce.

**Methods.** Data from 22,898 men and 28,541 women, aged 21 to 25 years, were collected from a national database in Taiwan. Five common oral diseases: dental caries, periodontitis, pulpitis, gingivitis, and stomatitis/aphthae were studied. Differences in the incidence of the five oral diseases in AR vs. non-AR, and AS vs. non-AS groups were compared. The incidence of the five oral diseases in men/ women, urban/country citizen, and high/low income groups was studied. The frequencies of clinical visits and impact of topical steroid use between the groups were also studied. The confounding factors included sex, socioeconomic status, urbanization, dentofacial anomalies, disease of salivary flow, diabetes mellitus, and esophageal reflux.

**Results.** The incidence and the frequencies of clinical visits for all five oral diseases were higher in those with AR than in the non-AR group after adjusting for confounding factors and AS. Similar observation was made for the AS group, without adjusting for AR. However, if AR was included for adjustment, no relationship was found between AS and oral diseases. In the AR group, those with higher incomes, and country residents had a high risk of developing oral disease. Intranasal steroids, rather than inhaled steroids, were also associated with oral diseases.

**Conclusion.** AR, rather than AS, may be associated with oral diseases in young adults.

## INTRODUCTION

Dental caries, periodontitis, pulpitis, gingivitis, and stomatitis/aphthae are common oral diseases (*Frencken et al., 2017*), which have a negative impact on quality of life and work performance. They also lead to increased medical costs. Understanding the risk factors for these conditions may help in their prevention and treatment. The oral diseases are

multi-factorial diseases. Some risk factors include oral pathogens, dry mouth, smoking habit, sugar consumption, and systemic diseases (*Hunter, 1988*; *Yousef, 2014*; *Roa & Del Sol, 2018*).

The association between AS, caries (*Alavaikko et al., 2011*) and periodontitis/gingivitis (*Moraschini, Calasans-Maia & Calasans-Maia, 2018*); and between AR, caries (*Wongkamhaeng, Poachanukoon & Koontongkaew, 2014*) and periodontitis (*Bakhshaee et al., 2017*) have been reported. However, the underlying mechanisms is different. Salivary micro-flora change due to mouth breathing (*Mummolo et al., 2018*). The effect of Interleukin (IL)-12 (*Ping et al., 2015*) has been reported in AR subjects. Decreased IgA levels in gingival tissue have been reported in AS subjects (*Ostergaard, 1997*). Oral micro-flora (*Sachs et al., 1993*; *Wongkamhaeng, Poachanukoon & Koontongkaew, 2014*) and drug prescription (*Tootla, Toumba & Duggal, 2004*; *Elad, Heisler & Shalit, 2006*) are two factors that influence oral diseases, and are different between AS and AR.

AR and AS often coexist. They are linked by epidemiologic, histologic, physiologic, and immunopathologic characteristics, and also by a common therapeutic approach (*Simons, 1999*). AS has been reported to be associated with both oral diseases (*Wongkamhaeng, Poachanukoon & Koontongkaew, 2014*; *Bakhshaee et al., 2017*) and AR. Compare to non-AR subjects, the frequencies of AS were higher in AR subjects (*Simons, 1999*). Therefore, we speculated that AS might be a confounder while studying the relationship between AR and oral diseases. AR has also been associated with oral diseases (*Alavaikko et al., 2011*; *Moraschini, Calasans-Maia & Calasans-Maia, 2018*) and AS. Compare to non-AS subjects, the frequencies of AR were higher in AS subjects (*Simons, 1999*). Therefore, we speculated that AR might be a confounder while studying the relationship between AS and oral diseases. Our previous study found that in children, AR rather than AS, is a risk factor for dental caries (*Chuang, Sun & Ku, 2018*). The confounder AR, brings the relationship between AS and caries into question in children. However, AS and AR, respectively, were not considered as confounders in the previous study on adults.

Studies on the association of AR with caries and periodontitis are few, and they show conflicting results (*Wongkamhaeng, Poachanukoon & Koontongkaew, 2014*; *Hung et al., 2016*; *Bakhshaee et al., 2017*; *Kim & Choi, 2018*). The association between AS and pulpitis and stomatitis; and between AR and pulpitis, gingivitis, and stomatitis have scarcely been studied. We performed a large population-based research study using the National Health Insurance Research Database (NHIRD) in Taiwan to establish the relationship among the five oral diseases (caries, periodontitis, pulpitis, gingivitis, and stomatitis/aphthae) and AS and AR in young adults. AR was adjusted for when studying the association between AS and oral disease, so we aimed to determine whether the association was induced by AR.

## MATERIALS & METHODS
### Database and data collection
The NHIRD was created by the National Health Research Institute (NHRI) in Taiwan (*Taiwan of Health, 1998*; *Cheng, 2003*). The NHRI randomly sampled a representative database of 1 million subjects in 2010 through systematic sampling, and this sample served

as our data source. The database provided information on patient identification, birth date, sex, diagnostic codes from the International Statistical Classification of Diseases and Related Health Problems (ICD)-9-CM, prescription drugs, medical care facilities, and other parameters. This medical information for each subject can be traced forward from year 1985 and backward to 2013.

## Criteria for AS, AR and oral diseases

Subjects born between 1985 and 1988 were selected randomly from the NHIRD. Their claims' data in the age group of 21 to 25 years were analyzed between 2005 and 2013. The subjects were divided into the groups: AR and non-AR; and AS and non-AS. The criteria for AR and AS included at least two diagnoses of AR (ICD-9-CM diagnostic code 477) or AS (ICD-9-CM diagnostic code 493) in five years between the ages of 21 and 25 years. Those with disease durations of AR and AS <180 days were excluded because AR and AS are chronic diseases. Five major oral diseases were selected for the study: dental caries (code 521.0), periodontitis (codes 523.3 and 523.4), pulpitis (code 522.0), gingivitis (codes 523.0 and 523.1), and stomatitis (code 528.0)/oral aphthae (code 528.2).

## Incidence of oral diseases

The subjects were followed up for five years (age 21–25 years old). Incidence of the five oral diseases in the AR versus non-AR groups and AS versus non-AS groups, was compared. More than one diagnosis of each oral disease recorded in the study period was defined as having the individual disease. Incidence of the five oral diseases for AR versus non-AR groups was compared using different demographic characteristics, which included male/female, urban/country resident, and high/low income. The magnitude of the relative risk (RRs) of oral diseases in each character was compared between AR and non-AR subjects.

## Clinical visit times for oral diseases

During the five years follow up, the mean clinical visit times for the five oral diseases in various groups were compared. Mean clinical visit times for three dental treatment methods: dental restoration, endodontics, and periodontitis treatment (surgical and non-surgical) were also compared to investigate the reason for clinical visit.

## Influence of use of inhaled steroids for AS and intranasal steroids for AR

AR subjects were divided into intranasal steroid group (who had used intranasal steroids during the study period) and non-intranasal steroid group (who had never used intranasal steroids during the study period). AS subjects were divided into inhaled steroid group (who had used inhaled steroids during the study period) and non-inhaled steroid group (who had never used inhaled steroids during the study period). Clinical visit times and treatments for oral disease were compared between the groups. Anatomical Therapeutic Chemical code R01AD was used for intranasal steroids, and R03AK and R03BA for inhaled steroids.

## Determining confounding factors

Factors that influence oral health include sex, socioeconomic status, urbanization, dentofacial anomalies (ICD-9-CM code 524), salivary flow diseases (ICD-9-CM codes 527 and 710.2), diabetes mellitus (DM) (ICD-9-CM code 250) and esophageal reflux (ICD-9-CM codes 530.11 and 530.81) (*Hunter, 1988*; *Yousef, 2014*; *Roa & Del Sol, 2018*). These factors were considered as risk factors and were adjusted for. AS was used as the confounding factor when investigating the association between AR and oral diseases, and AR was used as the confounding factor when investigating the association between AS and oral diseases.

Socioeconomic status was defined according to the occupation, wherein, the subjects were grouped into high-income (teacher or public official, company employee) and low-income (peasants, fishermen, others, low income or no fixed job) groups. Based on Liu's report (*Liu et al., 2006*), urbanization was grouped into seven levels: levels 1–2 for urban residents, and levels 3–7 for country residents.

## Statistical analysis

All analyses were performed using SAS version 9.1 for Windows (SAS Inc., Cary, NC, USA) and PASW Statistics 18 (IBM, Armonk, NY, USA). For the five oral diseases, we assessed model calibration with the Hosmer–Lemeshow goodness of fit test. The model was regarded as no goodness of fit in case of $p < 0.05$. Mantel-Haenszel chi-square tests stratified for different characteristics (sex, urbanization and income) were used to examine the potential effect modifiers among oral diseases and AR. A chi-square test was used to compare the frequencies of characteristics between groups. A modified Poisson regression analysis was used to obtain RRs for the incidence of oral diseases between the groups. Negative binominal regression was performed for assessing differences in the frequencies of oral disease and clinical treatment visits between the groups. Crude $p$ values and adjusted $p$ value for confounders (sex, socioeconomic status, urbanization, dentofacial anomalies, diseases of salivary flow, DM, esophageal reflux, AS, AR) were calculated. Two-sided $p$ values of <0.05 were defined as significant.

## RESULTS

### Demographic data, goodness of fit and potential effect modifiers

Demographic data are presented in Table 1. Total 51,439 subjects were recruited, of which 22,898 (44.5%) were men and 28,541 (55.5%) were women. Of these subjects, 7,884 (15.3%) fell into the AR group (3,247 (14.2%) men and 4,637 (16.2%) women), whereas, 1,232 (2.4%) fell into the AS group (569 (2.5%) men and 663 (2.3%) women). Hosmer–Lemeshow test $p$ values for the five oral diseases were as follows: caries (0.059), periodontitis (0.038), pulpitis (0.565), gingivitis (0.303), and aphthae/stomatitis (0.074). Sex, urbanization and income was the potential effect modifiers among caries, periodontitis and AR. Sex, and income was the potential effect modifiers among gingivitis and AR. Sex was the potential effect modifiers among gingivitis and AR.

**Table 1  Characteristics of AR vs non-AR and AS vs non-AS.**

|  | AR | Non-AR | Crude *p*-value | AS | Non-AS | Crude *p*-value |
|---|---|---|---|---|---|---|
|  | $n = 7,884$ (15.3%) | $n = 43,555$ (84.7%) |  | $n = 1,232$ (2.4%) | $n = 50,207$ (97.6%) |  |
| Sex |  |  |  |  |  |  |
|    Male | 3,247 (41.2%) | 19,651 (45.1%) | <0.001 | 569 (46.2%) | 22,329 (44.5%) | 0.232 |
|    Female | 4,637 (58.8%) | 23,904 (54.9%) |  | 663 (53.8%) | 27,878 (55.5%) |  |
| Socioeconomic status |  |  |  |  |  |  |
|    High income | 4,415 (56.0%) | 24,361 (55.9%) | 0.911 | 673 (54.6%) | 28,103 (56.0%) | 0.347 |
|    Low income | 3,469 (44.0%) | 19,194 (44.1%) |  | 559 (45.4%) | 22,104 (44.0%) |  |
| Urbanization |  |  |  |  |  |  |
|    Urban | 4,725 (59.9%) | 25,230 (57.9%) | <0.001 | 798 (64.8%) | 29,157 (58.1%) | <0.001 |
|    Country | 3,159 (40.1%) | 18,325 (42.1%) |  | 434 (35.2%) | 21,050 (41.9%) |  |
| Dentofacial anomalies | 1,466 (18.6%) | 5,748 (13.2%) | <0.001 | 191 (15.5%) | 7,023 (14.0%) | 0.130 |
| Disease of salivary flow | 92 (1.2%) | 240 (0.6%) | <0.001 | 12 (1.0%) | 320 (0.6%) | 0.145 |
| DM | 50 (0.6%) | 198 (0.5%) | 0.034 | 13 (1.1%) | 235 (0.5%) | 0.003 |
| Esophageal reflux | 295 (3.7%) | 700 (1.6%) | <0.001 | 65 (5.3%) | 930 (1.9%) | <0.001 |

**Notes.**

A chi-square test was used to compare the frequencies of characteristics between groups.

Abbreviations: AS, asthma; AR, allergic rhinitis; DM, diabetes mellitus.

## Incidence of oral diseases

The results for incidence of oral diseases are presented in Table 2. Incidence of the five oral diseases was higher in AR subjects as compared to non-AR subjects, and this difference was statistically significant. The statistically significant difference was still noted if AS was added for adjustment. The incidence of caries, periodontitis, gingivitis, and aphthae/stomatitis was higher in AS subjects as compared to non-AS subjects, and this difference was statistically significant. However, if AR was added for adjustment, all the differences became non-significant. The incidence of oral diseases between AR and non-AR subjects, considering the demographic characteristics, is presented in Table 3. Except for pulpitis in men, urban citizens and subjects with low income the incidence was all significantly higher in AR subjects.

To evaluate the magnitude of AR's influence on oral diseases, the RRs data for each demographic character and each oral disease was compared (Table 3). RRs (AR vs. non-AR) for dental caries was higher (1.17) in men than women (1.09). RRs (AR vs non-AR) of periodontitis, gingivitis and aphthae/periodontitis was higher in men than women. RRs (AR vs non-AR) of the five diseases was higher in AR country residents and AR subjects with high incomes than AR urban residents and AR subjects with low incomes.

## Clinical visits for oral diseases and treatment

Mean clinical visit times for the five oral diseases and three treatments were significantly higher in AR subjects during the study period (Table 4). The most common reason for clinical visit was restoration, followed by periodontitis treatment and endodontics (Table 4). The statistically significant difference was still noted if AS was added for adjustment. Except for endodontics treatment, mean clinical visit times for five oral diseases and two treatments

**Table 2  The incidence of oral disease in AR vs non-AR and AS vs non-AS subjects.**

|  | AR | Non-AR | *p*-value (a) | *p*-value (b) | *p*-value [RRs (95% CI)] (c) |
|---|---|---|---|---|---|
|  | *n* = 7,884 | *n* = 43,555 |  |  |  |
| Caries | 79.7% | 69.1% | <0.001 | <0.001 | <0.001 [1.12(1.11–1.14)] |
| Periodontitis | 71.5% | 59.9% | <0.001 | <0.001 | <0.001 [1.16(1.14–1.18)] |
| Pulpitis | 25.7% | 23.2% | <0.001 | <0.001 | <0.001 [1.08(1.04–1.13)] |
| Gingivitis | 60.5% | 49.5% | <0.001 | <0.001 | <0.001 [1.17(1.15–1.20)] |
| Aphthae/stomatitis | 29.7% | 19.8% | <0.001 | <0.001 | <0.001 [1.39(1.34–1.45)] |
|  | **AS** | **Non-AS** | ***p*-value (a)** | ***p*-value (b)** | ***p*-value [RRs (95% CI)] (d)** |
|  | *n* = 1,232 | *n* = 50,207 |  |  |  |
| Caries | 76.8% | 70.6% | <0.001 | <0.001 | 0.109 [1.03(0.99–1.06)] |
| Periodontitis | 67.8% | 61.5% | <0.001 | <0.001 | 0.290 [1.02(0.98–1.06)] |
| Pulpitis | 25.5% | 23.5% | 0.100 | 0.130 | 0.421 [1.04(0.94–1.15)] |
| Gingivitis | 57.7% | 51.0% | <0.001 | <0.001 | 0.115 [1.04(0.99–1.09)] |
| Aphthae/stomatitis | 27.2% | 21.2% | <0.001 | <0.001 | 0.104 [1.08(0.98–1.18)] |

**Notes.**

A modified Poisson regression was used to calculate the relative risks (RRs) and to adjust the confounders.

(a) Crude *p*-value.

(b) Adjusted by sex, socioeconomic status, urbanization, dentofacial anomalies, disease of salivary flow, DM, esophageal reflux.

(c) Adjusted by factors (b) plus AS.

(d) Adjusted by factors (b) plus AR.

Abbreviations: AS, asthma; AR, allergic rhinitis; CI, confidence interval; DM, diabetes mellitus.

were significantly higher in AS subjects; however, if AR was added for adjustment, the differences became non-significant.

## Association between inhaled steroids, intranasal steroids and oral diseases

The results for this association are presented in Table 5. In AR subjects, mean clinical visit times for the oral diseases except pulpitis, and treatment except endidontics, were significantly higher in those using intranasal steroids; and the statistically significant difference was still noted after adjustment for AS. In AS subjects, there was no significant association between mean clinical visit times for oral diseases and treatments of those who inhaled steroids and those who did not.

## DISCUSSION

Our study found that AR was associated with dental caries, periodontitis, pulpitis, gingivitis and aphthae/stomatitis. However, we did not find any association between AS and the five oral diseases after adjusting for AR. The clinical visit times for caries, periodontitis, gingivitis, aphthae/stomatitis, periodontitis, and restoration treatment times were higher in AR subjects using intranasal steroids. Also, in AR group, men, country residents and those with high income had higher incidence of oral diseases.

Studies on the relationship between AR and caries reveal conflicting results. Some studies found a positive association (*Bakhshaee et al., 2017*; *Chuang, Sun & Ku, 2018*),

**Table 3  The incidence of oral disease in AR vs non-AR, in different characters.**

|  | Caries | Periodontitis | Pulpitis | Gingivitis | Aphthae/stomatitis |
|---|---|---|---|---|---|
| Male |  |  |  |  |  |
| AR vs non-AR | 74.7%; 62.3% | 66.2%; 54.1% | 22.7%; 20.8% | 54.9%; 43.8% | 25.2%; 16.1% |
| RRs (95% CI) | 1.17(1.15–1.20) | 1.19(1.16–1.23) | 1.06(0.99–1.14) | 1.21(1.17–1.26) | 1.46(1.36-1.56) |
| *p*-value | <0.001 | <0.001 | 0.094 | <0.001 | <0.001 |
| Female |  |  |  |  |  |
| AR vs non-AR | 83.2%; 74.7% | 75.3%; 64.6% | 27.8%; 25.1% | 64.5%; 54.2% | 32.9%; 22.9% |
| RRs (95% CI) | 1.09(1.08–1.11) | 1.14(1.12–1.16) | 1.10(1.04–1.16) | 1.15(1.12–1.18) | 1.36(1.30–1.43) |
| *p*-value | <0.001 | <0.001 | <0.001 | <0.001 | <0.001 |
| Urban |  |  |  |  |  |
| AR vs non-AR | 79.9%; 70.7% | 72.1%; 61.9% | 24.8%; 23.2% | 60.9%; 51.3% | 29.1%; 20.0% |
| RRs (95% CI) | 1.11(1.09–1.12) | 1.14(1.12–1.17) | 1.05(0.99–1.11) | 1.15(1.12–1.18) | 1.37(1.30–1.44) |
| *p*-value | <0.001 | <0.001 | 0.081 | <0.001 | <0.001 |
| Country |  |  |  |  |  |
| AR vs non-AR | 79.4%; 66.9% | 70.7%; 57.3% | 27.0%; 23.1% | 60.1%; 47.0% | 30.6%; 19.9% |
| RRs (95% CI) | 1.15(1.12–1.17) | 1.18(1.15–1.21) | 1.13(1.06–1.21) | 1.22(1.18–1.26) | 1.42(1.34–1.51) |
| *p*-value | <0.001 | <0.001 | <0.001 | <0.001 | <0.001 |
| High income |  |  |  |  |  |
| AR vs non-AR | 80.5%; 68.5% | 72.4%; 58.5% | 24.4%; 21.6% | 61.4%; 50.0% | 29.5%; 19.6% |
| RRs (95% CI) | 1.14(1.12–1.16) | 1.18(1.16–1.21) | 1.11(1.05–1.18) | 1.18(1.15–1.21) | 1.41(1.33–1.48) |
| *p*-value | <0.001 | <0.001 | <0.001 | <0.001 | <0.001 |
| Low income |  |  |  |  |  |
| AR vs non-AR | 78.7%; 69.9% | 70.4%; 60.4% | 27.4%; 25.2% | 59.4%; 48.9% | 29.8%; 20.1% |
| RRs (95% CI) | 1.09(1.07–1.12) | 1.13(1.10–1.16) | 1.06(0.99–1.12) | 1.17(1.13–1.21) | 1.37(1.29–1.46) |
| *p*-value | <0.001 | <0.001 | 0.075 | <0.001 | <0.001 |

**Notes.**

A modified Poisson regression was used to calculate the relative risks (RRs) and *p*-value.

*p*-value: adjusted by sex, socioeconomic status, urbanization, dentofacial anomalies, disease of salivary flow DM, esophageal reflux and AS.

Abbreviations: AS, asthma; AR, allergic rhinitis; CI, confidence interval; DM, diabetes mellitus.

while others found no association (*Tanaka et al., 2008*; *Wongkamhaeng, Poachanukoon & Koontongkaew, 2014*). No studies have been reported in adults. The relationship between AR and periodontitis is inconclusive, because both positive (*Hung et al., 2016*) and inverse (*Kim & Choi, 2018*) associations have been reported. To the best of our knowledge, an association between AR and gingivitis, pulpitis, and stomatitis/aphthae has not been reported previously.

Our study provides evidence that AR is associated with five oral diseases (caries, periodontitis, pulpitis, gingivitis, and stomatitis/aphthae). The increased prevalence rate may imply that AR influences development of oral diseases, while increased clinical visit times mean that AR might increase the severity of oral diseases. Restoration and endodontics treatment are used for caries, periodontitis, and pulpitis; and increased treatment visit times in AR subjects make the association more certain. Among the AR subjects, men, country residents and those with high income had higher RRs for oral diseases. This might suggest

**Table 4 Mean clinical visit times (over 5 years) for oral disease and treatments, for AR vs. non-AR; and AS vs. non-AS subjects.**

|  | AR | Non-AR | Ratio | *p*-value (a) | *p*-value (b) | *p*-value [95% CI] (c) |
|---|---|---|---|---|---|---|
| Caries | 3.91 | 3.08 | 1.27 | <0.001 | <0.001 | <0.001 [1.19–1.27] |
| Periodontitis | 2.16 | 1.61 | 1.34 | <0.001 | <0.001 | <0.001 [1.25–1.33] |
| Pulpitis | 0.55 | 0.48 | 1.15 | <0.001 | <0.001 | <0.001 [1.05–1.18] |
| Gingivitis | 1.42 | 1.04 | 1.37 | <0.001 | <0.001 | <0.001 [1.26–1.35] |
| Aphthae/stomatitis | 0.60 | 0.35 | 1.71 | <0.001 | <0.001 | <0.001 [1.49–1.66] |
| Periodontitis treatment | 6.70 | 5.08 | 1.32 | <0.001 | <0.001 | <0.001 [1.25–1.30] |
| Restoration | 7.45 | 6.05 | 1.23 | <0.001 | <0.001 | <0.001 [1.17–1.23] |
| Endodontics | 1.79 | 1.62 | 1.10 | <0.001 | 0.004 | 0.009 [1.02–1.13] |
|  | **AS** | **Non-AS** | **Ratio** | ***p*-value (a)** | ***p*-value (b)** | ***p*-value [95% CI] (d)** |
| Caries | 3.60 | 3.20 | 1.13 | <0.001 | 0.001 | 0.461 [0.96–1.10] |
| Periodontitis | 1.94 | 1.69 | 1.15 | <0.001 | <0.001 | 0.541 [0.95–1.10] |
| Pulpitis | 0.58 | 0.49 | 1.18 | 0.014 | 0.020 | 0.096 [0.98–1.29] |
| Gingivitis | 1.32 | 1.09 | 1.21 | <0.001 | <0.001 | 0.095 [0.99–1.15] |
| Aphthae/stomatitis | 0.52 | 0.39 | 1.33 | <0.001 | <0.001 | 0.210 [0.95–1.25] |
| Periodontitis treatment | 6.09 | 5.31 | 1.15 | <0.001 | <0.001 | 0.374 [0.97–10.8] |
| Restoration | 7.03 | 6.24 | 1.13 | <0.001 | 0.001 | 0.329 [0.97–1.11] |
| Endodontics | 1.82 | 1.64 | 1.11 | 0.102 | 0.097 | 0.243 [0.95–1.22] |

**Notes.**

Negative binomial regression was used to calculate the frequencies and to adjust the confounders.

(a) Crude *p*-value.

(b) Adjusted by sex, socioeconomic status, urbanization, dentofacial anomalies, disease of salivary flow, DM, esophageal reflux.

(c) Adjusted by factors (b) plus AS, with 95% CI.

(d) Adjusted by factors (b) plus AR, with 95% CI.

Abbreviations: AS, asthma; AR, allergic rhinitis; CI, confidence interval; DM, diabetes mellitus.

that men who are country residents with a high income should pay more attention to their oral health if they also have AR.

There have been more studies investigating the association between AS and caries, periodontitis and gingivitis, than on the association with AR. Many studies have reported a positive association between AS and caries (*Alavaikko et al., 2011*), and periodontitis and gingivitis (*Moraschini, Calasans-Maia & Calasans-Maia, 2018*). However, most of the studies did not adjust for AR, which might be an important confounder. In our study, before adjusting for AR, incidence for four oral diseases (Table 1), clinical visit times for five oral disease and three oral treatments was significantly higher in AS group than the non-AS group. However, after adjusting for AR, the differences became non-significant. So, we concluded that AS is not associated with oral disease. AR might be a confounding factor associated with both AS and oral diseases. Inhaled steroids have been reported to increase the incidence of caries, periodontitis, gingivitis, and oral ulcers, with change in oral pH, local deposition of steroids in the oral cavity, and their effect on oral mucosa (*Alavaikko et al., 2011*; *Bozejac et al., 2017*). The study of intranasal steroids and oral diseases is limited. We speculate that local deposition of steroids due to postnasal dripping might induce the same oral diseases that inhaled steroids do. In our study, three oral diseases and two oral treatment visit times were significantly associated with the use of intranasal steroids, but

**Table 5** Mean clinical visit times (over 5 years) for oral disease and treatments for AR subjects who had used intranasal steroids (38.6) vs. those who never had (61.4%), and AS subjects who had used inhaled steroids (55.4%) vs. those who never had (44.6%).

| AR with nasal steroid | yes | no | Ratio | p-value (a) | p-value (b) | p-value [95% CI] (c) |
|---|---|---|---|---|---|---|
| Caries | 4.18 | 3.74 | 1.12 | <0.001 | <0.001 | <0.001 [1.07–1.18] |
| Periodontitis | 2.28 | 2.09 | 1.09 | <0.001 | <0.001 | <0.001 [1.04–1.15] |
| Pulpitis | 0.57 | 0.54 | 1.06 | 0.279 | 0.257 | 0.281 [0.95–1.18] |
| Gingivitis | 1.53 | 1.35 | 1.13 | <0.001 | <0.001 | <0.001 [1.07–1.19] |
| Aphthae/stomatitis | 0.64 | 0.57 | 1.12 | 0.021 | 0.032 | 0.035 [1.01-1.22] |
| Periodontitis treatment | 7.08 | 6.46 | 1.10 | <0.001 | <0.001 | <0.001 [1.06–1.14] |
| Restoration | 7.83 | 7.21 | 1.09 | <0.001 | <0.001 | <0.001 [1.04–1.14] |
| Endodontics | 1.85 | 1.75 | 1.06 | 0.242 | 0.183 | 0.209 [0.97–1.17] |

| AS with inhaled steroid | yes | no | Ratio | p-value (a) | p-value (b) | p-value [95% CI] (d) |
|---|---|---|---|---|---|---|
| Caries | 3.44 | 3.73 | 0.92 | 0.208 | 0.307 | 0.195 [0.81–1.04] |
| Periodontitis | 1.99 | 1.90 | 1.05 | 0.489 | 0.564 | 0.752 [0.90–1.16] |
| Pulpitis | 0.58 | 0.57 | 1.02 | 0.928 | 0.767 | 0.735 [0.80–1.38] |
| Gingivitis | 1.25 | 1.39 | 0.90 | 0.168 | 0.153 | 0.069 [0.76–1.01] |
| Aphthae/stomatitis | 0.48 | 0.55 | 0.87 | 0.280 | 0.505 | 0.172 [0.65–1.08] |
| Periodontitis treatment | 6.06 | 6.12 | 0.99 | 0.830 | 0.805 | 0.411 [0.87–1.06] |
| Restoration | 6.97 | 7.07 | 0.99 | 0.825 | 0.943 | 0.623 [0.86–1.10] |
| Endodontics | 1.89 | 1.76 | 1.07 | 0.580 | 0.491 | 0.545 [0.85–1.37] |

**Notes.**

Negative binomial regression was used to calculate the frequencies and to adjust the confounders.

(a) Crude p-value.
(b) Adjusted by sex, socioeconomic status, urbanization, dentofacial anomalies, disease of salivary flow, DM, esophageal reflux.
(c) Adjusted by factors (b) plus AS, with 95% CI.
(d) Adjusted by factors (b) plus AR, with 95% CI.
Abbreviations: AS, asthma; AR, allergic rhinitis; CI, confidence interval; DM, diabetes mellitus.

no association was noted between inhaled steroids and oral diseases (Table 4). Intranasal steroids are more expensive than inhaled steroids in Taiwan, so these drugs are prescribed if patients have more severe AR symptoms. More studies are needed to explain our findings.

The results of several studies could explain the association between AS, AR, and oral diseases. At the mechanistic level, a decreased saliva secretion rate has been reported in both AS (*Lenander-Lumikari et al., 1998*) and AR (*Elad, Heisler & Shalit, 2006*) subjects. Dehydration of the gingiva due to mouth breathing in AR subjects may also be a contributing factor for gingivitis. At the pharmacological level, antihistamines (*Elad, Heisler & Shalit, 2006*) and inhaled $\beta_2$-agonists (*Tootla, Toumba & Duggal, 2004*) could decrease salivary flow. At the microbiological level, AS (*Sachs et al., 1993*) and AR (*Wongkamhaeng, Poachanukoon & Koontongkaew, 2014*) subjects have been found to have different oral micro-flora than the control subjects. At the immunological level, decreased IgA levels in gingival tissue have been reported in allergic disease (*Ostergaard, 1997*), and IgA is a first-line defense immunoglobulin for mucosa, and plays a role in restricting periodontal disease. Another study indicated that IL-12 is associated with AR (*Ping et al., 2015*); IL-12 may also be related to the pathogenesis of periodontal disease (*Tsai et al., 2005*).

Our study has several strengths. First, other studies have assessed AR or AS from past history, rather than being an existing condition at the time of study. AR and AS subjects in

our study were in the active stage of allergic disease with more than two diagnoses within the study period. Studying the association between oral diseases and AR and AS during the active stage of allergic diseases is more pertinent. Second, by knowing the frequencies of clinical visits and treatment times, we could understand the influence of allergic disease on the severity of oral diseases, medical treatment, and medical costs.

There are limitations to our study. First, there are no laboratory data such as oral pathogenic bacteria, IgA in oral samples/saliva and mucus. Second, there are no data on smoking habit, sugar consumption, engagement with the healthcare system, personal oral hygiene, and oral drugs. Third, the Hosmer–Lemeshow test was used to evaluate the model's calibration. Caries, pulpitis, gingivitis and aphthae/stomatitis showed a good fit, but periodontitis did not. Future study for the association between AR and periodontitis is necessary.

## CONCLUSIONS

The present study provides evidence that AR might be associated with caries, periodontitis, pulpitis, gingivitis, and stomatitis/aphthae in young adults. Based on increased clinical visit times for the five diseases, it can be speculated that AR also increases the severity of these five oral diseases. Contrary to findings in other studies, there is no association between AS and the five oral diseases. Any association between AS and oral diseases previously found might be due to AR.

Men who live in the countryside and have a high income should pay more attention to oral hygiene if they also have AR because incidence of the oral diseases was found to be higher in these demographic categories. Intranasal steroids used in AR, rather than inhaled steroids used in AS, are associated with the development of oral diseases.

In order to prevent and treat oral diseases, simultaneous treatment of AR is important. Considering the effects of AR in the elucidation of the etiology of oral diseases is a new direction of study. However more studies are required to ensure whether AR is the cause of oral diseases. More biological research and more epidemiological data on the relationship between AR, AS and oral diseases are necessary in order to gain clarity.

### Funding
The authors received no funding for this work.

### Competing Interests
The authors declare there are no competing interests.

### Author Contributions
- Sai-Wai Ho performed the experiments, analyzed the data, contributed reagents/materials/analysis tools, prepared figures and/or tables, authored or reviewed drafts of the paper.
- Ko-Huang Lue approved the final draft.

- Min-Sho Ku conceived and designed the experiments, performed the experiments, analyzed the data, contributed reagents/materials/analysis tools, prepared figures and/or tables, authored or reviewed drafts of the paper, approved the final draft.

**Data Availability**

Data is available at National Health Insurance Research Database, Taiwan (http://nhird.nhri.org.tw/en/index.html). The data utilized in this study cannot be made available due to the "Personal Information Protection Act" executed by Taiwan's government, starting from 2012. Requests for data can be sent as a formal proposal to the NHIRD (http://nhird.nhri.org.tw) or by email to nhird@nhri.org.tw.

If readers are interested in the study, you can email the corresponding author for details at a129184@yahoo.com.tw.

**Supplemental Information**

Supplemental information for this article can be found online at http://dx.doi.org/10.7717/peerj.7643#supplemental-information.

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
