# Peer review of "Allergic rhinitis, rather than asthma, might be associated with dental caries, periodontitis, and other oral diseases in adults"

_PeerJ, doi:10.7717/peerj.7643_

## Round 0.1 · original submission · Major Revisions

The authors must satisfy the critiques from reviewers

[]

·

Basic reporting

Some of the text needs rewriting to correct errors (e.g. Lines 39, 40, etc.) and other parts of the manuscript could, I think, be expressed more simply and clearly (e.g. Lines 43–44, 47–49, 52, etc.). I suggest careful proofreading throughout.

It is unclear what a “co-confounder” (Lines 40, 74–75, and elsewhere) is—Google provides only 18 hits for this term and most of these appear to be spurious matches.

The term “rate” is usually reserved for cases where the denominator is non-trivial (e.g. per year). The use on Line 45 and elsewhere is the trivial case of per person and normally this would be described as prevalence or incidence as appropriate instead.

When stating that adjustment was made for “confounding factors” (e.g. Line 50), it is important that the reader has a list of these variables in mind so they can judge whether this is likely to be adequate or inadequate for the purpose.

The study is not longitudinal (Lines 82 and 237–239) in any meaningful sense—while the data were collected over time, the analyses as described here are strictly cross-sectional in nature.

A traditional Table 1 for patient information is needed to present their demographic characteristics.

Tables 1–2: Note that tildes are usually used to indicate “approximately” rather than a range of values.

Tables 3–4: 95% CIs should also be provided here along with effects

Tables 1–4: Results should be presented for both sets of models, not just p-values.

All tables should indicate the exact statistical models used to produce the results presented there. There were some parts of the statistical methods that did not seem to clearly match what was in the tables.

While the raw data is supplied, it needs a data dictionary. For example, it is unclear simply by looking at the data whether sex=1 or sex=2 represents males (I appreciate that this question can be answered by comparing the frequencies in the data with those in the manuscript, but I think that this should be apparent from the data alone). A data dictionary for all 40 columns needs to be available. I suggest using a second worksheet in the workbook for this purpose. It might be a consequence of the lack of a data dictionary, but I couldn’t see a variable for the presence/absence of aphthae/stomatitis.

Experimental design

While the manuscript frequently mentions adjustment for confounders, there is no discussion of an underlying causal model that has informed the design of the study. Without careful consideration of this, perhaps through a directed acyclic graph, it is difficult to know what the actual research question being investigated is. While the section named “Determining confounding factors” (Line 129) discussed what variables were included, there is only one reference in this section and that is for how urban/rural levels were created. All confounders and other variables included need to be justified, and references should be provided as part of this whenever possible. This process will include identifying potential confounders that are not available, which will lead into the limitations later on (Lines 248–250). Note also that at least some of the variables as described here will clearly leave residual confounding (SES was dichotomised, for example, and obesity appears to be based on a diagnosis rather than BMI) if they are indeed confounders. Can the authors explain why they collapsed these variables to fewer levels? It is unclear to me why pregnancy is grouped with sex and obesity in this text (Lines 138–139). Sex and obesity could, potentially and depending on the causal model, be confounders whereas pregnancy does not seem likely to me to be a confounder. Finally, could you please explain in the manuscript how AS is a confounder for the AR->oral disease association, and how AR is a confounder for the AS->oral disease association?

Loosely related to this, “multivariate” (Lines 146, 150, 151, and possibly elsewhere) should be “multivariable” (the former refers to multiple outcomes, the latter to multiple predictor variables).

If logistic regression models are going to be used for multivariable models, why use Chi-squared tests for the univariable models (Lines 145–146)? The mathematical assumptions are slightly different, but the p-values will be similar either way and by starting with univariable logistic regression, there is a smoother transition through the analyses. Even better, though, in my opinion, would be to use modified Poisson regression (https://doi.org/10.1093/aje/kwh090), or some other approach, to model relative risks rather than to estimate odds ratios which will inevitably be confused by some readers as being RRs (which they will not approximate well given the common outcomes investigated here). For the comparisons of frequencies (Lines 149–150), t-tests will be inappropriate due to the inevitable skew in these outcomes and their discrete nature. Poisson, or negative binomial, regression might be appropriate here depending on the data.

What model diagnostics were performed for the statistical models? The reader should be informed about these in the manuscript so that they can judge the internal validity of the findings. The exact models are not clear in all cases, but there is fairly consistent evidence of misfit in the models using Hosmer-Lemeshow’s test (while this test is not without its problems, some form of diagnostics are still needed).

Were any interactions/effect modifiers considered plausible and investigated? Interactions between AR and AS seem inherently plausible here and this is statistically significant for caries (p=0.003) with and without further adjustment. Further, by adding interactions, Table 2 will be much more useful with a direct test of effect modification included. At the moment, the reader is unable to determine simply from the information presented here whether or not effects differ by sex, urbanity, or income in most all cases.

Validity of the findings

It is traditional to begin the discussion with a short summary of the results, and I suggest adding such a paragraph at Line 190.

There is considerable over-interpretation of the study findings in my opinion. Because these are cross-sectional data, and I think there are missing confounders (see above and note your own discussion points in the manuscript), I would prefer the language to reflect this uncertainty. For example, describing AR is a risk factor in the title is claiming an actual causal association which I do not think can be justified here. AR might be a diagnostic/prognostic marker, but I cannot see a strong mechanistic explanation for the claimed causal association sufficient to reduce concerns that the results may simply reflect unmodelled confounding. The conclusion in the abstract (Line 56), for example, is an absolute statement that simply has not been justified here. This issue continues throughout the manuscript—causal claims can only be justified where all important confounding variables have been accounted for, which is an inherently difficult requirement in observational research. The discussion claims on Lines 203–204 are also overstated and the causal claim on Lines 252–253 cannot be justified in my view.

The potential confounding effect of AR on the AS->oral health association is still not clear to me despite being stated as a fact in the Discussion (Lines 208–209). How exactly would AR confound this association? Later in the Discussion, you soften this to “might be a [confounder]” (Lines 212–213).

While I appreciate the possible mechanisms suggested, I still feel that omitted variable bias (smoking, sugar consumption, engagement with the healthcare system, and other health experiences and behaviours could potentially be adequate explanations in themselves for the observed results) is a potentially important threat to the study’s internal validity. This isn’t to dispute that further research might not be worthwhile, but I feel that the observed associations, if they persisted, would have to be a speculative point and only after the issues with the statistical models are resolved.

The sample size (Lines 239–240) is not in itself a strength except insofar as it affects precision and stability of estimates. Related to the former point, I think more interpretation of the magnitudes of effects and 95% CI limits is needed. A major limitation would appear to be the lack of temporal sequencing in the exposure and outcome. If I have understood the data extraction process correctly, this was not a requirement, but please do correct me otherwise.

Additional comments

While the study has some interesting aspects, there are some important points to address, including a) providing a causal model to underlie the statistical analyses, b) addressing the limitations of the statistical modelling and/or reporting, and c) the over-interpretation of the study findings, particular in terms of causality.

·

Basic reporting

In the manuscript, Allergic rhinitis, rather than asthma, is a risk factor for dental caries, periodontitis, and other oral diseases in adults, Min-Sho Ku et al study association between oral disease and allergic rhinitis versus asthma in young adults.

Strengths: The manuscript is well written and discussion is appropriate. The data collected is robust in terms of number of subjects and timeframe of analysis. The statistically significant association of AR with oral disease is noted. The study does take into account age, sex, lifestyle (country v urban) and clinical visits.
Weakness:
The scientific rationale for studying oral disease’s association with AR or asthma has not been well described in the introduction.. They have a previous paper on similar topic in children yet they need to introduce why allergic diseases are relevant to oral diseases.
The association between higher income groups having more oral disease/AR is surprising. They need to discuss this better.
The number of clinical visit needs to be classified; meaning purpose of clinical visit should be taken into consideration.
Most importantly, the authors themselves state their weakness on lack of oral hygiene data. I think without that data, it is not reliable to make any clear associations between AR-oral diseases versus AS-oral diseases.
The level of IgA in oral samples/saliva and mucus should be taken into consideration as well.

Overall, the manuscript needs improvement on the weaknesses to be published.

Experimental design

Experimental design is mostly appropriate; Consideration of reason for clinical visit should be mentioned/taken into account. Oral hygiene must be accounted for. Use of mouth wash, brushing teeth (once v twice daily) etc are a major player in all oral diseases and should be considered.

Validity of the findings

no comment

Additional comments

Make the introduction better with details on why studying the connection between oral disease and allergic diseases are scientifically and biologically rational and important. State clearly the differences between asthma and allergic rhinitis.

---

## Round 0.2 · Minor Revisions

The authors should address the comments from the reviewer.

·

Basic reporting

The line numbers I previously listed as having language issues were only for the first few such lines. In the abstract, there remain language errors on Lines 38 (adult), 39 (remain), 44 (disease), 45 (list is not divided into sub-lists but refers to these categories in the final items), 46 (this is awkward), 47 (incidence), 50 (missing definite article), 51 (high), and 53 (missing word). I’m not listing the line numbers here to indicate specific errors and corrections, and I could in theory keep listing line numbers throughout the manuscript, but rather to support my suggestion that the manuscript would benefit from careful proofreading throughout.

I still feel that the confounders adjusted for should be made clear in the abstract on Line 47 as it is difficult to interpret the findings without knowing at least the types of variables adjusted for here. Note also that these are just some of potential confounding factors and the statement here reads somewhat more definite than that to me.

Thank you for adding Table 1. I suggest looking at making this seven columns (possibly in landscape) so you can avoid repeating the first column for both variables. Also, I suggest “<0.001” for very small p-values (as you use in Tables 2 and 3, etc.) rather than “0.000” (the p-value is not actually zero). The capitalisation in Table 1 is inconsistent in the first column (e.g. “disease…” and “Obesity”). Finally, the table should indicate the test(s) used in the table in notes under the table so the reader can quickly find this information (and this should also be described in the statistical methods). This is what I meant by my previous comment: “All tables should indicate the exact statistical models used to produce the results presented there.” (Which you have done for Tables 2, 3, 4, and 5.)

In Table 2, the effects from the modified Poisson regression are relative risks and the reader will find it easier to follow with this indicated as “RR” rather than “IRR”. Same in Table 3 and throughout the text (e.g. Line 121 and 161).

In Table 2, it remains unclear to the reader which p-value is associated with the relative risks and CIs. Personally, I’d rather see all three sets of RRs, 95% CIs, and p-values so the effects of adjustment are more conspicuous; but at the very least, the reader should not wonder which p-value is attached to the reported effects. The same issue arises in Table 4, which uses a different layout, now separating the effect and CI into different columns rather than having them in a single column.

While you have added data for aphthae/stomatitis and improved the names of variables, the data continues to need a data dictionary. For example, as noted previously, it is unclear to me simply by looking at the data whether sex=1 or sex=2 represents males (I appreciate that this question can be answered by comparing the frequencies in the data with those in the manuscript, but I think that this should be apparent from the data alone). The same point applies to other variables, such as urbanization and income. A data dictionary for all columns needs to be available. I suggest using a second worksheet in the workbook for this purpose.

Experimental design

I feel that the causal model underlying the analyses still needs attention despite the additional text on this. Something being a risk factor for poor oral health is not sufficient for it to be a confounder. For example, if obesity is a consequence, more than as a result, of asthma (through a lack of exercise) and a consequence of poor oral health (through food choices given difficulties eating), this would make obesity a collider rather than a confounder for the association between asthma and oral health, i.e. a variable that must not be included in a model to estimate the total effect of asthma on oral health. On the other hand, if asthma led to obesity (which is also the case) and obesity led to poor food choices and so worse oral health, obesity would be on the causal pathway and again should not be included in a model to estimate the total effect of asthma on oral health. As I said last time, it is also plausible that obesity is indeed a confounder of this association, but this would need to be justified in the manuscript. I suggest starting with the definition of a confounder and establishing that this is more plausible than not for each of these variables. Epidemiology is seldom straightforward!

I’d asked if you could please explain in the manuscript how rhinitis is a confounder for the asthma->oral health association, and how rhinitis is a confounder for the asthma->oral health association. I cannot see any direct justification for your speculation that “AS might be an confounder when study the relationship between AR and oral disease, and AR might be an confounder when study the relationship between AS and oral disease.” (Lines 76–78). The second paragraph in the introduction doesn’t, for me, provide a reason to speculate this—two different mechanisms would not suggest that they might mutually confound one another in the relationship with poor oral health. I appreciate that you use “might be” on Line 238 for this possible confounding when looking at the literature, but I still cannot see precisely what would lead you to suspect this particular confounding in the first place. Again, I suggest starting with the formal definition of a confounder and considering these three variables in that context. I expect that answering this question will require expanding the causal model well beyond these three variables, with all the modelled variables here ultimately included, perhaps along with some that are not available here.

Thank you for adding information about model diagnostics, but I am confused as to how Hosmer-Lemeshow was used with Poisson regression. This is normally an option for logistic regression. Why did you use negative binomial regression (Line 162) rather than Poisson regression? What approach did you use for deciding between these options?

Validity of the findings

There are possibly some minor errors in the reported results (but note that I have only looked at a small number of results and not all presented values). The upper CI for caries and rhinitis in Table 2 should be 1.14 not 1.37 I think. I get p=0.101 (not 0.100) for the p-value for pulpitis in model (a) asthma. I get p=0.112 for gingivitis (not 0.118) for model (d). Finally, I get 0.99 for the lower CI limit for aphthae/stomatitis rather than 0.98 (my value was 0.985 so I suspect this is rounding). It is possible, but seems unlikely for such convention models, to get different results using different packages or, more likely, I’m misunderstanding some aspect of the model, so if your results are indeed as included in the table, I apologise and we can put this down to my error or a SAS/Stata difference in how these models are implemented.

While you mention looking for effect modification (and note that Breslow Day and MH both test homogeneity of odds ratios rather than relative risks and the questions here concern the fully adjusted models), you say there was no evidence for this, which, if true, would make the stratified presentation in Table 3 unnecessary. I’m not entirely sure how to reconcile Lines 174–175 with your later statements, e.g. Lines 189–193 where you note that some strata have higher RRs than others. However, based on my quick analyses, there is indeed statistically significant evidence of effect modification in the adjusted models looking at rhinitis for caries by sex (Wald p<0.001 for the rhinitis-by-sex interaction), caries by income (p<0.001), caries by urbanisation (p=0.002), periodontitis by sex (p=0.002), periodontitis by income (p=0.007), periodontitis by urbanisation (p=0.009), gingivitis by sex (p=0.004), gingivitis by urbanisation (p=0.003), and aphthae/stomatitis by sex (p=0.015). For the first of these, you can informally see this by the non-overlapping CIs (1.15–1.20 and 1.08–1.11) which indicates p<0.01 in this case, with a non-surprising p-value (p=0.015) for the very last given this rule and the fact that the CI limits just overlap.

I’m not clear on how “First, the data were collected over time (5 years).” (Line 266) is a study strength. For strengths and limitations, I think you need to tease out how each is a strength or weakness. I’m not sure that the lack of data on sugar consumption needs to be mentioned twice on Line 275. What are the “laboratory results” on Line 276 referring to?

Additional comments

I think that the topic is worthwhile and the research question asked seem important in guiding future research. The manuscript is much improved from last time and with some very careful attention to the causal model, including effect modification, and the English, I think a useful article is possible here.

---

## Round 0.3 · accepted · Accept

The authors have satisfied the comments from the reviewers